# The Application of Electrical Parameters to Reflect the Hydration Process of Cement Paste with Rice Husk Ash

**DOI:** 10.3390/ma12172815

**Published:** 2019-09-02

**Authors:** Hui Wang, Lingxuan Hu, Peng Cao, Baoxin Luo, Jing Tang, Feiting Shi, Jinyan Yu, Hongjie Li, Kaikai Jin

**Affiliations:** 1School of Civil and Environmental Engineering, Ningbo University, Ningbo 315000, China; 2School of Civil Engineering, Harbin Institute of Technology, Harbin 150090, China; 3College of Architecture and Civil Engineering, Beijing University of Technology, Beijing 100124, China; 4Civil Engineering Department, Yancheng Institute of Technology, Yancheng 224051, China

**Keywords:** electrical resistivity, AC impedance spectroscopy, cement paste, rice husk ash, equivalent circuit

## Abstract

This paper aims to study the electrical parameters (electrical resistivity and alternating current (AC) impedance spectroscopy) of cement paste with rice husk ash (RHA). The water to cement (Mass ratio of water to cement (w/c)) ratios of the paste in this study varied from 0.4 to 0.5. The mass ratio of rice husk ash in each w/c ratio of specimens ranged from 0% to 15% by t mass of cement. Scanning electron microscopy (SEM) and X-ray diffraction (XRD) were used to determine the microstructures of specimens. Moreover, the slump flow and plastic viscosity of fresh paste were determined. The results indicated that with the increasing dosage of RHA, the fluidity decreased, while the plastic viscosity increased. Meanwhile, a high w/c ratio led to a low plastic viscosity and high slump flow. The electrical resistivity of RHA cement paste gradually ascended with the increasing curing period. The conduction of specimens intricately changed by mixing RHA, a reasonable equivalent circuit was selected to describe the conduction mechanism by AC impedance spectroscopy. Additionally, the results of XRD and SEM showed that RHA could effectively promote the hydration process as well as decrease the size and number of cracks in hardened cement paste.

## 1. Introduction 

Rice husk ash (RHA) is produced from agricultural waste (rice husk) [1,2,3,4]. The main component of RHA contains a large amount of amorphous silica. Therefore, like silica fume, RHA is a kind of active mineral admixture [5]. Zhuang et al. pointed out that RHA is effective to decrease the porosity and Ca(OH)_2_ in cement paste [6]. Moreover, as described in some literatures, RHA possesses a very high pozzolanic reactivity comparable with that of silica fume (SF) [7,8,9]. RHA has been proven by scholars to replace silica fume for achieving high strength/performance concrete [10,11]. 

Van et al. reported that the addition of RHA can reduce free water in high strength/performance concrete, thus improving its mechanical properties [12]. Additionally, RHA is able to decrease the pore size and numbers in concrete, thus enhancing the compressive strength and impermeability [13]. Moreover, RHA was considered as an internal curing agent to decrease autogenous shrinkage [14]. However, RHA may decrease the fluidity of fresh concrete. The influences of RHA on the rheological properties of fresh concrete and the rheological mechanism were unknown. 

The hydration of cement material is a complicated physical and chemical process [15]. This process controls the microstructure and macro performance of cement matrix [16]. Many literatures on studying hydration mechanism have been published [17]. Generally, some researchers analyzed the hydration and hardening mechanisms by means of scanning electron microscopy (SEM), X-ray diffraction (XRD), and Fourier transform infrared spectroscopy (FT-IR) [18,19]. It is known that continuous monitoring of hydration process information may not be obtained from the above methods. In addition, the preparation environment when drying samples for microscopy observations and analysis to characterize hydration process will also affect the test results. Aiming at these problems, some scholars have begun to focus on nondestructive testing methods, electrical testing methods, microwave energy methods, and ultrasound detection methods. These methods are the most commonly used testing methods [20,21,22]. Compared with the methods mentioned above, nondestructive testing (NDT) is undoubtedly a better way to analyze the complex hydration process of cement materials.

Recently, electrical properties of cement-based materials were determined to reflect the service performance of concrete, e.g., the hydration process etc. Wang et al. pointed out that cement mortar with carbon nanofibers was able to sense freeze-thaw damage of materials, environmental temperatures, applied stress, strain etc. by sensing the evolution of the specimen’s electrical resistivity [23]. The self-sensing cement composite with conductive fibers is called intrinsic self-sensing concrete [24]. Moreover, Han et al. found that the electrochemical impedance spectroscopy (EIS) and equivalent circuits are able to analyze the conductive mechanisms of cement-based composite with conductive fillers [25]. However, few researchers focus on the conductive performance of plain RHA cement paste with different rheological properties and curing ages. Obviously, little attention was paid to electrical properties of cement-based materials that reflected the hydration process of RHA cement matrix [26].

This paper studies the influence of rice husk ash content and curing ages on the rheological properties (slump flow and plastic viscosity) and electrical parameters (electrical resistance and alternating current (AC) impedance spectroscopy) of cement paste. Mass ratio of RHA varied from 0% to 15% by the total mass of bind materials. In this research, water-cement (w/c) ratios were 0.4, 0.45, and 0.5. SEM technology has greatly helped to elucidate the microstructures, and the XRD method was applied to determine the composition of specimens with RHA. 

## 2. Experiment

### 2.1. Raw Materials

The 42.5 ordinary Portland cement (P.O 42.5) provided by Yancheng Conch Cement Co., Ltd. (Yancheng, China) was used as the binder material in this research. The fluidity of fresh paste was adjusted by a polycarboxylate-based, high-range water-reducing agent. The RHA was fabricated by burning origin rice husk in an airtight high temperature furnace. The furnace filled with rice husk was first heated at a rate of 10 °C to 500 °C for 2 h, and then the calcined residue was ground in a vibrating mill for 15 min. The particle size distribution and chemical compound were the same as in Reference [27]. The particle passing percentage and chemical composition of cement and RHA are shown in Table 1 and Table 2. In Table 2, “R_2_O” means “Na_2_O + K_2_O”. The particle passing percentage of cementitious materials was provided by Yancheng Conch Cement Co., Ltd. The method for testing the particle passing percentage of cementitious materials was obtained according to the testing method for the particle size of cement. Additionally, the chemical composition of cement and RHA were chosen in a previous study [27].

### 2.2. Mixing Proportion

Table 3 shows the mixing proportions of all groups of the paste. In this experiment, the w/c ratios were 0.4, 0.45, and 0.5. In each group of specimens, the dosage of RHA varied from 0% to 15%.

### 2.3. Specimens Preparation and Measurement

Cement and water were mixed uniformly in the NJ-160A cement paste mixer and stirred at a speed of 140 rpm for 2 min. Next, the mixture was mixed again at a speed of around 285 rpm for another 2 min. After that, the slump flow of the prepared fresh paste was measured by using a steel cone with a bottom diameter of 60 mm, a top diameter of 36 mm and a height of 60 mm. Additionally, an NXS-11B rotating viscometer manufactured by Chengdu Leadership Instruments Co., Ltd. (Chengdu, China) was used to determine the plastic viscosity of the paste. To measure the electrical resistance, specimens with a size of 20 mm × 30 mm × 35 mm were fabricated and then the electrical resistances were tested by a TH2810D LCR digital electric bridge (Changzhou Tonghui Co., Ltd., Changzhou, China) with a frequency ranging from 100 Hz to 10 kHz. The tests were conducted at an AC voltage of 1 V and frequency of 10 kHz with the embedded two-pole layout method, as shown in Figure 1a. The electrodes were made of corrosion resistant AISI 316L stainless steel mesh with a size of 2.0 cm × 3.5 cm and a thickness of 0.8 mm. The AC electrical resistance along the longitudinal axis direction of the cube was determined. Three specimens were measured for each group. The electrochemical impedance spectroscopy was tested by an electrochemical workstation (Shanghai Chen Hua Instrument Co., LTD, Shanghai, China) as shown in Figure 1b. The microstructure and crystal types of hydration products were determined by JSM-6360LV scanning electron microscope (Japan electron optics laboratory, Tokyo, Japan) and D8 ADVANCE X-ray diffractometer (Bruker Corp., Tokyo, Japan) respectively. Before the experiments, the samples were dried in a vacuum drying oven (Shanghai Yi Heng Instrument Co., LTD DZF-6020, Shanghai, China) at 60 °C. When SEM observations were carried out all samples were metallized with gold.

## 3. Results and Discussion

### 3.1. Rheological Performance of Cement Paste 

Figure 2 shows the variation of slump flow and plastic viscosity with the increasing dosage of RHA. The shear rate for plastic viscosity determination was 8 rpm. As shown in Figure 2, more RHA resulted in a lower slump flow and higher plastic viscosity. Moreover, higher w/c ratios presented higher slump flow and lower plastic viscosity. The increasing content of RHA decreased the slump flow and increased plastic viscosity because the incorporation of RHA could adsorb some free water in fresh paste that had resulted in decreasing the fluidity and increasing the viscosity of fresh paste [12]. However, higher w/c ratios led to higher fluidity and lower viscosity, this was attributed to the fact that cement paste with higher w/c ratios contains more free water, therefore the flowing property was improved, and the viscosity was decreased by higher w/c ratios [28].

### 3.2. Electrical Resistivity

Figure 3, Figure 4 and Figure 5 show the electrical resistivity of specimens with RHA content ranging from 0% to 15%. As shown in Figure 3, Figure 4 and Figure 5, the resistivity of all specimens increased with the increasing curing time because the increasing hydration of the cement paste might decrease the free conductive ions in the cement paste thus resulting in decreasing the conductivity [28–31]. Therefore, the resistivity of specimens increased with increasing curing ages. At the curing ages from 3 h to 24 h, the electrical resistivity of the cement paste with RHA from 0% to 10% linearly increased with the curing ages. However, for the specimens with 15% RHA content, the electrical resistivity of all specimens rarely changed with the curing ages. Moreover, the electrical resistivity of specimens with w/c ratios of 0.4 and 0.45 decreased slightly with RHA content. This might be attributed to the fact that RHA is a pozzolanic material, hence, the portlandite first formed and then reacted with RHA, and thus the setting time and hardening were delayed with the increase in RHA content [21]. However, when the curing ages ranged from 336 h to 672 h, the electrical resistivity of specimens with w/c ratios of 0.4 and 0.45 increased slightly with RHA content, because RHA is effective to diminish conductive free ions by accelerating hydration and decreasing the fluidity of fresh paste. Above all, the conductive performance of specimens with RHA is a complex process. The addition of RHA was effective to decrease the fluidity of the fresh paste and reduce the free water in the cement paste, thus decreasing the conductive ions of pore solution and increasing the resistivity of specimens. The retarding action of RHA in cement paste might decrease the resistivity of specimens [21]. In this study, w/c ratios had little influence on the resistivity of specimens. This might be attributed to some complicated reasons that will be studied in the future.

### 3.3. Electrochemical Impedance Spectroscopy (EIS)

The electrical resistances of the cement paste with different w/c ratios are shown in Figure 6, Figure 7 and Figure 8. EIS and corresponding equivalent circuits were used to analyze the mechanism of conductive properties. The imaginary and real components of the electrochemical impedance spectroscopy represent the electrical reactance and resistance, respectively [25]. It can be seen that the imaginary and real components of all groups increase linearly with real components as frequency decreases from left to right. The Chi-squared is less than or equal to 2.053 × 10^−3^. This verifies that the selected equivalent circuits are reasonable for describing the conductive pathways in Figure 6, Figure 7 and Figure 8. When the water to cement ratio (Mass ratio of water to cement) was 0.40, the growth rate of imaginary components with real components was accelerated by the increased curing ages. However, when the water cement ratios were 0.45 and 0.50, the growth rates of the imaginary and real components were random. The internal causes will be studied in the future. Meanwhile, as shown in Figure 6, Figure 7 and Figure 8, the increase of the curing ages was effective to increase the real components, which meant that the electrical resistances of specimens increased as well. The incorporation of RHA can decrease the real components at the initial curing ages (3–24 h) but increase the real components at the long-term curing ages (1–28 d). This behavior was described and analyzed in Part 3.2. 

Figure 9 shows the corresponding equivalent circuits of specimens with RHA. The circular section can be represented by the parallel of resistance and capacitance elements. The initial resistance (R_s_) corresponds to the contact resistance between specimen and electrodes. Then, the following series connected electrical components represented shunt-wound resistance (R_1_) and electrical capacitance (C_1_) of the pore solution. A conductive pathway was formed by the pore solution in the cement matrix. The addition of RHA can decrease the fluidity and enhance the hydration thus blocking the conductive channels in specimens. Moreover, due to the electrical capacitances of specimens, the specimens present strong zero drift phenomena in the electrical performance.

### 3.4. Microscopic Analysis

Figure 10 shows the XRD patterns of specimens after a 28-day standard curing. From Figure 10 it can be seen that the diffraction peaks of 3CaO·SiO_2_ (C_3_S), 2CaO·SiO_2_ (C_2_S), and SiO_2_ are significantly strong, which indicates good crystallinity [32,33]. Results indicated that all specimens were mainly constituted of unhydrated cement particles such as 3CaO·SiO_2_ (C_3_S), 2CaO·SiO_2_ (C_2_S), SiO_2_, and hydration products like calcium hydroxide (CH) and hydrated calcium silicate (C-S-H). Figure 10 shows that the height of diffraction peaks of SiO_2_ increased with the increase of RHA, whereas the height of diffraction peaks of CH and C_3_S decreased with the increase of RHA. These results can be attributed to the high RHA content leading to a low fraction of SiO_2_ in the cement paste. Moreover, the increasing dosage of RHA might have accelerated the hydration process of the cement paste, which led to the decrease of CH and C_3_S [17,18,19].

Figure 11 shows the scanning electron microscope (SEM) photos of specimens. The curing ages of all specimens were 28 days. As shown in Figure 11, Ca(OH)_2_ crystals can be observed obviously when RHA is not added to the paste. Many flocky hydration products exist in specimens when they contain RHA. Moreover, cracks in specimens decreased with the increasing dosage of RHA. Additionally, the decreasing w/c ratios led to reduce the numbers and size of crack in specimens. Results indicate that RHA was effective to improve the compactness and decrease the cracks in specimens. Previous research pointed out that the amorphous silica in RHA reacted with calcium hydroxide forming C-S-H. Moreover, as described in Sensale’s research, the particle size of RHA was less than 45 µm, as being the nucleation point due to hydration products [34]. Therefore, RHA was able to improve the compactness of the cement paste.

## 4. Conclusions

Based on this study, the following conclusions can be drawn.

(1)The slump flow of fresh cement paste was reduced, and the plastic viscosity was increased by the increasing addition of RHA and lower w/c ratios.(2)The resistivity of all specimens increased with the increasing curing ages. The addition of RHA had a complex influence on the conduction of specimens, due to its influence on the number of electrical free ions and the retarding action in cement paste. The equivalent circuits consisted of three resistor elements and three capacitance elements. Resistors or capacitances were in series connection respectively. A parallel circuit was more reasonable between each capacitance and resistor.(3)RHA was effective to improve the compactness of the cement paste. Moreover, the hydration products (CH) were decreased by the increasing addition of RHA.

## Figures and Tables

**Figure 1 materials-12-02815-f001:**
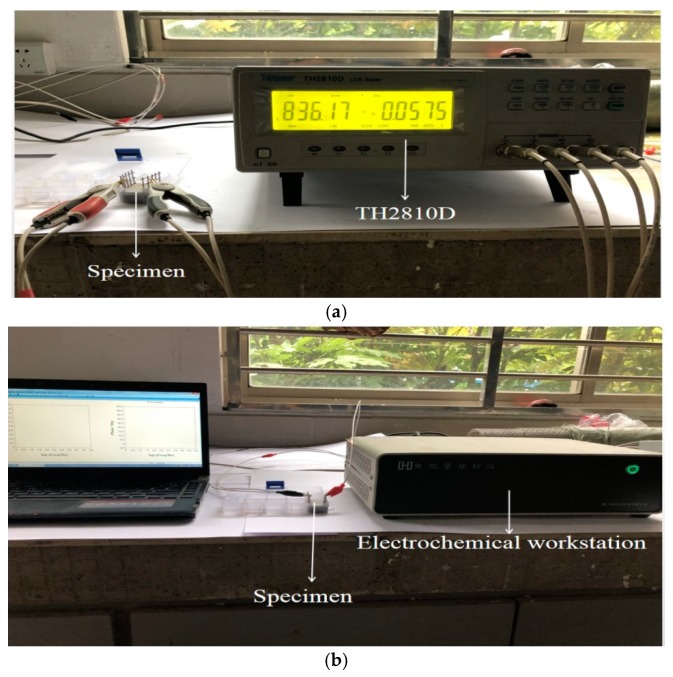
Electrical parameters measurement: (**a**) Electrical resistance measurement; (**b**) Alternating current (AC) impedance spectroscopy.

**Figure 2 materials-12-02815-f002:**
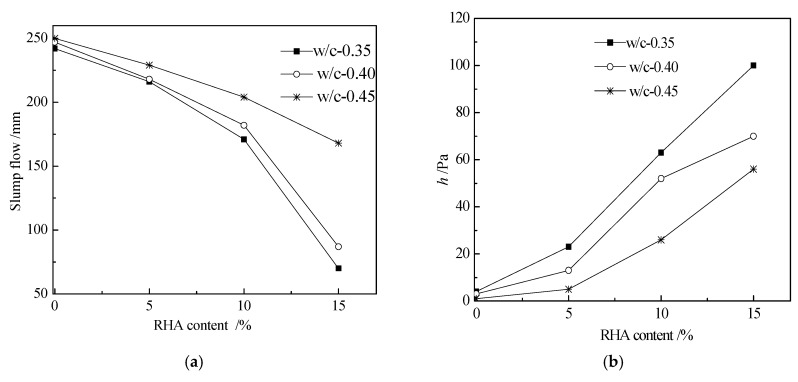
Slump flow and viscosity of fresh paste with different dosage of Rice husk ash (RHA): (**a**) Slump flow; (**b**) Viscosity.

**Figure 3 materials-12-02815-f003:**
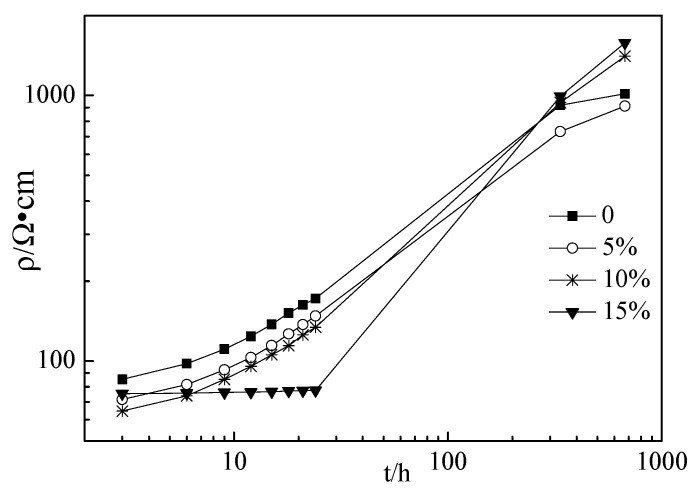
Electrical resistivity of specimens with w/c ratio of 0.40.

**Figure 4 materials-12-02815-f004:**
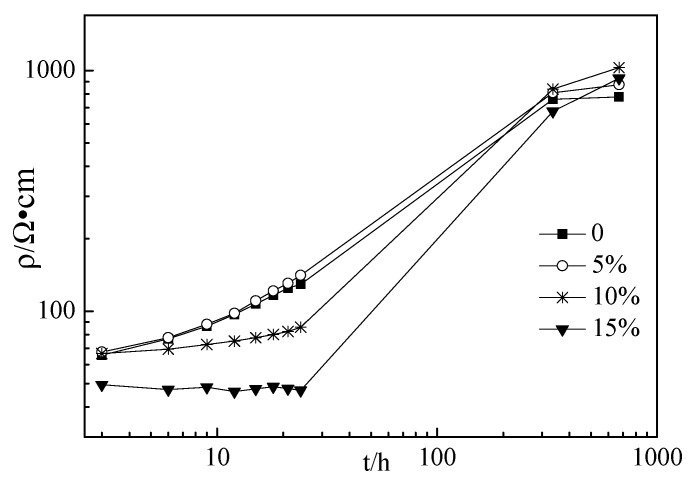
Electrical resistivity of specimens with the w/c ratio of 0.45.

**Figure 5 materials-12-02815-f005:**
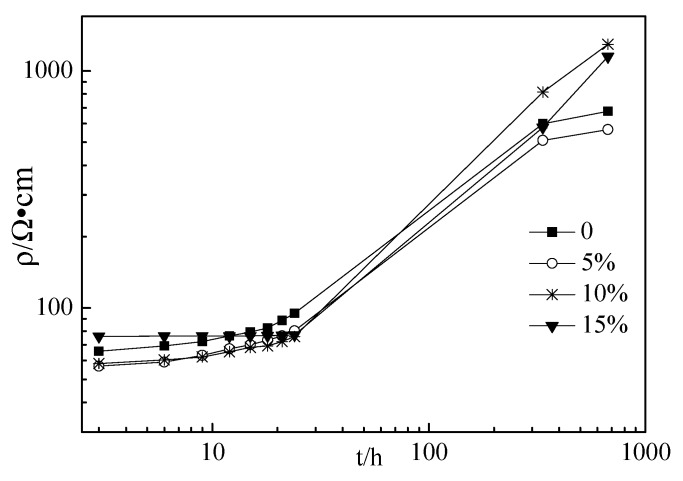
Electrical resistivity of specimens with the w/c ratio of 0.50.

**Figure 6 materials-12-02815-f006:**
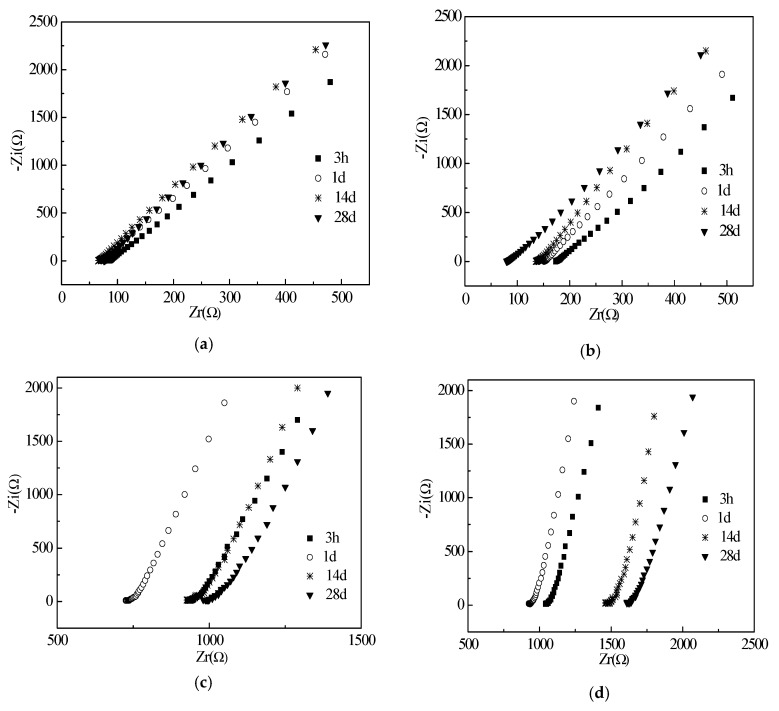
Electrochemical impedance spectroscopy (EIS) curves of specimens with w/c ratio of 0.40: (**a**) RHA-1; (**b**) RHA-2; (**c**) RHA-3; (**d**) RHA-4.

**Figure 7 materials-12-02815-f007:**
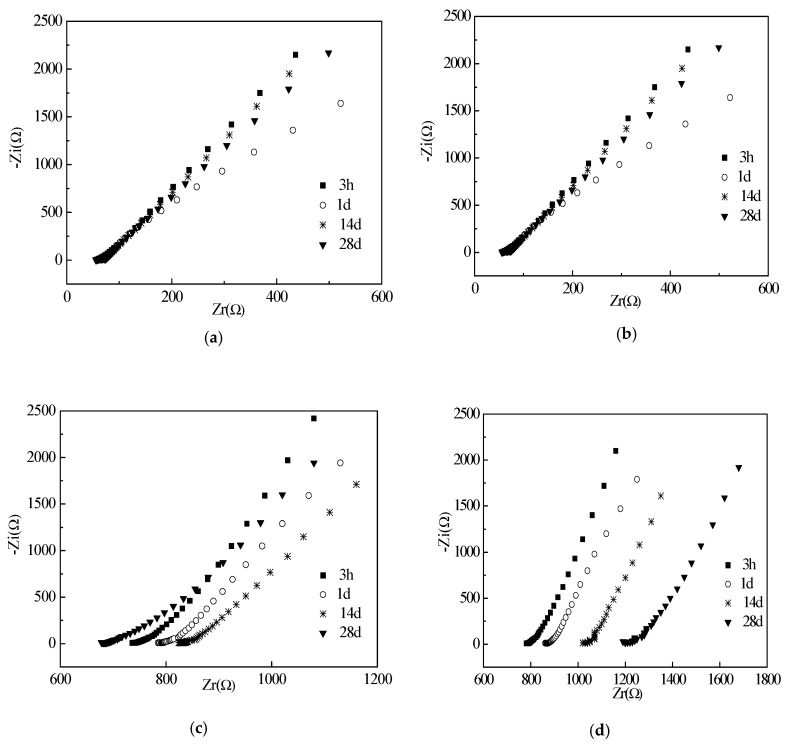
EIS curves of specimens with w/c ratio of 0.45: (**a**) RHA-5; (**b**) RHA-6; (**c**) RHA-7; (**d**) RHA-8.

**Figure 8 materials-12-02815-f008:**
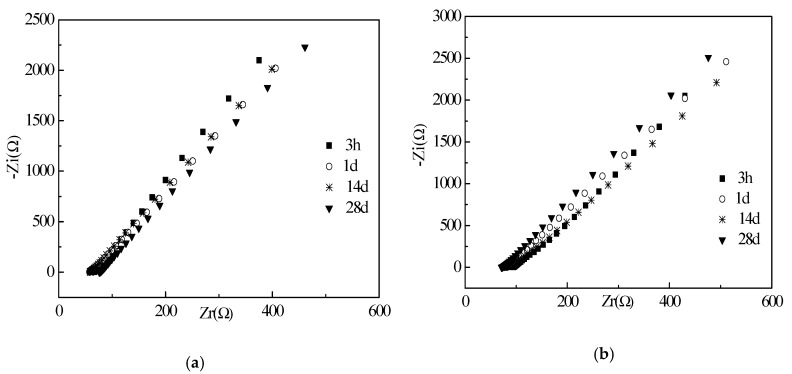
EIS curves of specimens with w/c ratio of 0.50: (**a**) RHA-10; (**b**) RHA-11; (**c**) RHA-12; (**d**) RHA-12.

**Figure 9 materials-12-02815-f009:**
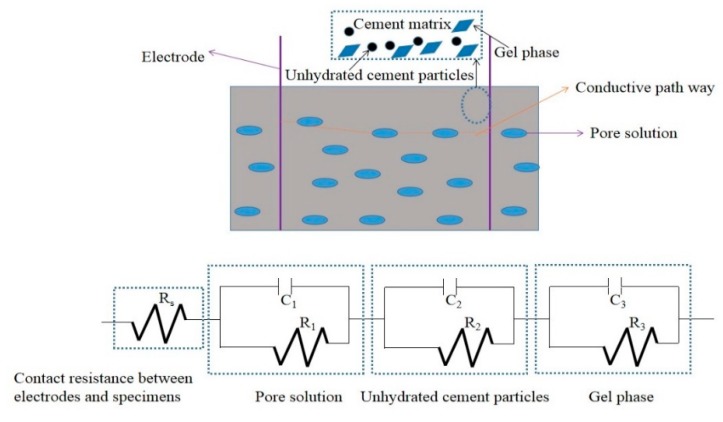
Corresponding equivalent circuits of specimens with RHA.

**Figure 10 materials-12-02815-f010:**
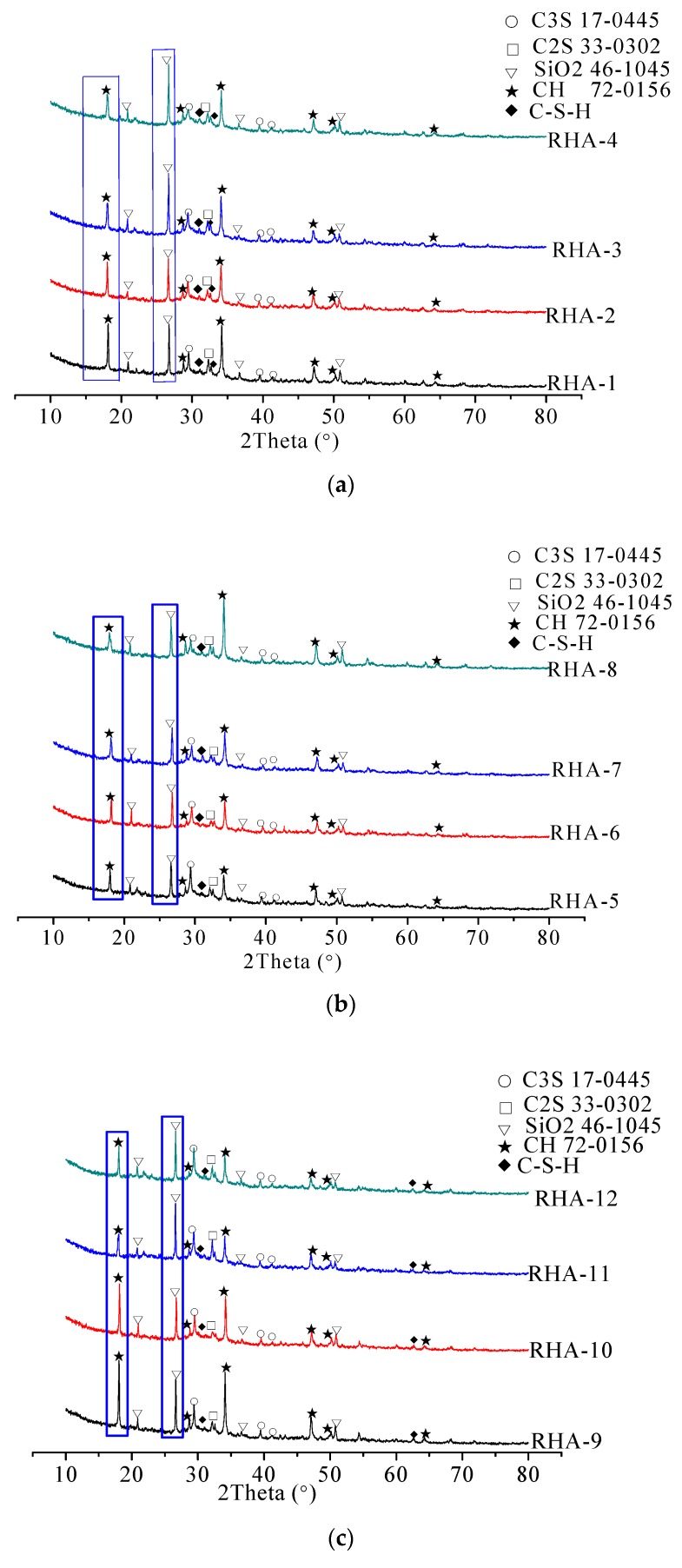
XRD spectra of specimens with RHA: (**a**) w/c ratio of 0.40; (**b**) w/c ratio of 0.45; (**c**) w/c ratio of 0.50.

**Figure 11 materials-12-02815-f011:**
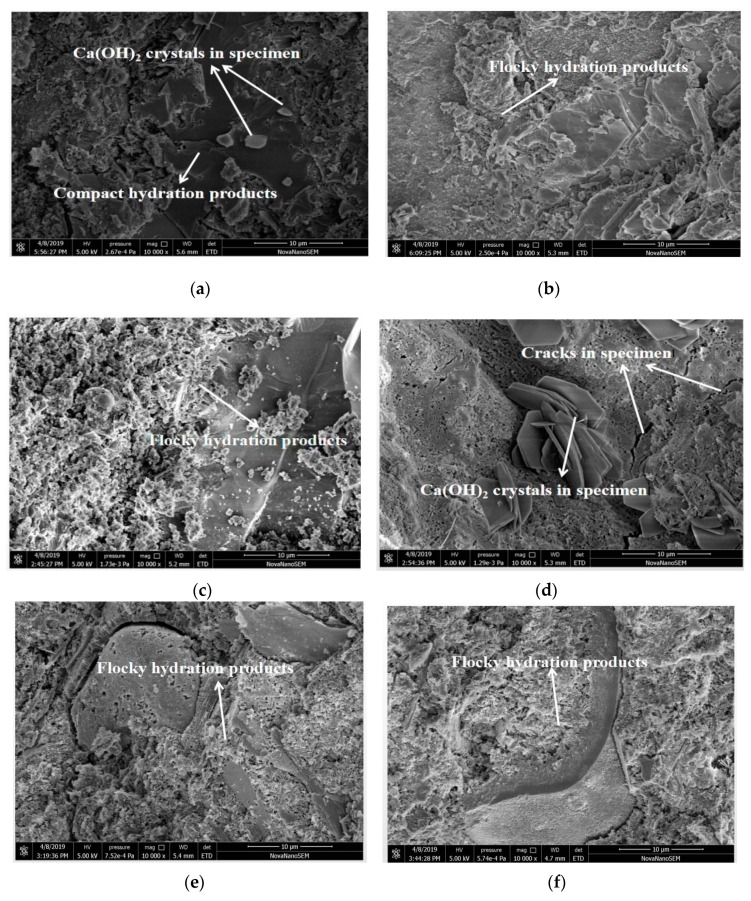
SEM micrographs of specimens with RHA: (**a**) RHA-1 (10000); (**b**) RHA-2 (10000); (**c**) RHA-3 (10000); (**d**) RHA-4 (10000); (**e**) RHA-5 (10000); (**f**) RHA-6 (10000); (**g**) RHA-7(10000); (**h**) RHA-8 (10000); (**i**) RHA-9 (10000); (**j**) RHA-10 (10000); (**k**) RHA-11 (10000); (**l**) RHA-12 (10000).

**Table 1 materials-12-02815-t001:** Particle passing percentage of the cementitious materials.

	Particle Size/um	0.3	0.6	1	4	8	16	32	64
Types									
Cement (P.O 42.5)		0	0.33	2.66	15.01	28.77	46.64	72.73	93.59
RHA		0	0.58	6.84	18.32	32.14	51.62	78.45	96.32

**Table 2 materials-12-02815-t002:** Chemical composition of the cementitious materials.

Types	Chemical Composition/%								
	SiO_2_	Al_2_O_3_	Fe_2_O_3_	MgO	CaO	SO_3_	R_2_O	MnO	H_2_O
Cement (P.O 42.5)	20.86	5.47	3.94	1.73	62.23	2.66	0.48	0	0
RHA	91.56	0.19	0.17	0.65	1.07	0.47	3.92	0.74	0.23

**Table 3 materials-12-02815-t003:** Mixing proportion of fresh cement paste.

Samples	Cement	Water	RHA	Water Reducing Agent
RHA-1	100	40	0	0.6
RHA-2	95	40	5	0.6
RHA-3	90	40	10	0.6
RHA-4	85	40	15	0.6
RHA-5	100	45	0	0.6
RHA-6	95	45	5	0.6
RHA-7	90	45	10	0.6
RHA-8	85	45	15	0.6
RHA-9	100	50	0	0.6
RHA-10	95	50	5	0.6
RHA-11	90	50	10	0.6
RHA-12	85	50	15	0.6

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
