# Peer review of "The Application of Electrical Parameters to Reflect the Hydration Process of Cement Paste with Rice Husk Ash"

_materials, 2019, doi:10.3390/ma12172815_

Round 1

Reviewer 1 Report

It's an interesting manuscript which can be published after addressing the following comments. 1. Please explain line 34, how high strength was achieved? The following study showed that supplementary binder reduce the porosity to improve the eprformance. "Thermal properties and residual strength after high temperature exposure of cement mortar using ferronickel slag aggregate" 2. Please explain how alkali-aggregate reaction was mitigated? you can use the following paper as a guideline which showed that pozzolanic reaction was effective to do so. "Mitigation of the potential alkali-silica reaction of ferronickel slag (FNS) aggregate by using ground FNS as a supplementary cementitious material" 3. What is the unit of contents of table 3? 4. The legends of fig. 3 and 4 is unclear. 5. Fig. 11 brightness need to be adjusted. 6. Line 251 "RHA could improve the hydration degree." not a valid conclusion as it was not measured.

Author Response

Paper Title: Electrical parameter applied in reflecting the hydration process of cement paste with rice husk ash

Manuscript Number: materials-552037

Referee’s comments for Author

Reviewer #1

It's an interesting manuscript which can be published after addressing the following comments.

1.  Please explain line 34, how high strength was achieved?The following study showed that supplementary binder reduce the porosity to improve the peprformance. "Thermal properties and residual strength after high temperature exposure of cement mortar using ferronickel slag aggregate"

-----Thank you for your comment. As described in some publications, RHA possesses a very high pozzolanic reactivity comparable with that of silica fume (SF) [7-9]. RHA was proved to replace silica fume for achieving high strength/performance concrete by scholars [10, 11].

2.  Please explain how alkali-aggregate reaction was mitigated? you can use the following paper as a guideline which showed that pozzolanic reaction was effective to do so. "Mitigation of the potential alkali-silica reaction of ferronickel slag (FNS) aggregate by using ground FNS as a supplementary cementitious material"

-----This is a very good point. I had revised as the reviewer suggested.

3. What is the unit of contents of table 3?

-----This is a very good point. The contents of Table 3 possess no unit, the numbers in Table 3 reflects the proportion of each content.

4. The legends of fig. 3 and 4 is unclear.

-----Thank you. The legends of fig. 3 and 4 is revised as all reviewers suggested.

5. Fig. 11 brightness need to be adjusted.

-----This is a good point. The brightness of Fig. 11 was adjusted.

6. Line 251 "RHA could improve the hydration degree." not a valid conclusion as it was not measured. 

-----Thank you for your suggestion. This conclusion was revised.

Reviewer 2 Report

All comments presented in file: see attachment

Author Response

Reviewer #2

1. All comments presented in file: see attachment

-----Thank you for your comments. All issues in this attachment was revised  according to the comments.

2. As a whole, the manuscript is very poorly and casually written. The overall presentation of the manuscript is vague. The novelty part is also missing in the present manuscript.

-----This is a good point. The manuscript was revised thoroughly according to all reviews comments. The novelty of this paper is the electrical parameters (electrical resistance and AC impedance spectroscopy) applied in reflecting the hydration process of cement paste with rice husk ash. The novelty part is added in the introduction part.

3. What was the principle to use to design an experiment? How do they complement / confirm each other?

-----Thank you for your suggestion. The principle to use to design the experiment was the electrical of cement paste containing RHA in different curing ages. The internal mechanism for development of electrical and microscopic properties were added in the paper.

4. In 3.2 and 3.3 section, one of parameter is hydration time. Moreover, in part 3.4 is a constant. Why? 

-----This is a good point. In 3.2 and 3.3 section, influence of RHA content, curing ages, water-cement ratios and parameters on the electrical and rheological performances of cement paste were studied. However, in part 3.4 the authors just want to study the influence of RHA contents and water-cement ratios on the microscopic performances of cement paste, therefore, the influencing curing ages were ignored.

5. In 3.4 part.:- The quality of XRD curve is poor, used etalons; d-spacing; pdf number (add it in text).
- All peaks must be identified.
- You misrepresent the XRD results:
Hydration products: portlandite (with calcite), CSH, ettringite: (see below:)

-----Thank you for your suggestion. Just as review suggested all peaks were added, errors of XRD results were corrected.

6. The intensity of primary peak of cement dependent of additive concentration in the samples. This has not been rated.

-----Thank you for your suggestion. Errors were corrected as suggested.

7. Quantitative composition of products: The description of XRD analysis should be combined with thermal analysis results.

-----This is a good point. The description of XRD analysis in this paper was preliminary qualitative analysis of hydration process, further research will be carried out in the future.

Reviewer 3 Report

Dear Authors,

the topic of your paper is worth of interest. However, this article can't be accepted in its present form, in my opinion. 

Specifically, please check:

* Line 16, Abstract: check the exact name of the technique, please. In the paper you didn't performed any compositional analysis...

* Lines 38, 39: please,  check:

1) the sentence: does RHA favor or prevent alkali aggregate reaction?

2) the reference [10] which doesn't describe alkali aggregate reaction in cementitious materials prepared with RHA.

* Lines 49, 50: not clear: check the sentence, please.

* Lines 59, 60: not clear: check the sentence, please.

* Line 79: acronym (SP) should be explicited to help Readers, even if obvious + Cite the type of SP and the producer...

* Lines 82,83: How did you determine the particle size distribution? With a laser granulometer? If yes, please mention it and the model you used...

* Table 1: what are SL and SF??? You never cited them before! Check, please.

* Line 97: Please detail: XRD instrument used and experimental conditions. Idem for SEM observations: did you dry the samples? metallised them?

* Line 102: which is the viscometer manufacturer, even if obvious? Which spindle did you use?

* Line 104: which is the  manufacturer of TH2810D instrument?

* Line 121: what was the shear rate for plastic viscosity determination?

* Line 144: How can there be a polarization resistance if you worked with an AC voltage at 10 kHz? Check, please. It may be due to the fact that RHA is a pozzolanic material and you have to let first portlandite to form and then to react with RHA. Thus, setting time and hardening are delayed with the increase in RHA content.

* Lines 150-152: As previously written: how can there be a polarization resistance if you worked with an AC voltage at 10 kHz? Did you use Vicat needle to determine setting time? If not, you should determine it.

* Line 198: Please, mention the JCPDS card numbers you used for XRD peaks indexation.

* Line 201: How can C4AF content, which is due to OPC, increase when RHA content increases? Check, please.

* Lines 201-203: C3S decrease with increasing addition of RHA  is due  to the fact that you have 15 wt% less cement in RHA4 sample respect to RHA1...

* Figure 10: Is the non-indexed peak below 30° due to CSH? Please, check

* Line 222: what does it mean that the microstructure improves? This is not clear.

* Line 222: the decrease of cracking is rather surprising, as usually, RHA increases cracking: Ameri et al., Optimum rice husk ash content and bacterial concentration in self-compacting concrete, Construction and Building Materials 222 (2019) 796–813. You should explain this.

* Figure 11: are compact parts in RHA8 and RHA12 samples RHA grains?

* Line 246: How can the sample polarize when doing AC measurements?

* Line 250: what does it mean that the microstructure improves? This is not clear.

+ see the attached pdf file for other minor revisions.

Best regards.

Reviewer1

Author Response

Reviewer #3

Dear Authors,

the topic of your paper is worth of interest. However, this article can't be accepted in its present form, in my opinion. 

Specifically, please check:

* Line 16, Abstract: check the exact name of the technique, please. In the paper you didn't performed any compositional analysis...

-----Thank you for your suggestion. The name of the technique was checked and the errors were corrected.

* Lines 38, 39: please,  check:

1) the sentence: does RHA favor or prevent alkali aggregate reaction?

2) the reference [10] which doesn't describe alkali aggregate reaction in cementitious materials prepared with RHA.

-----This is a good point. This was an error and the point was revised.

* Lines 49, 50: not clear: check the sentence, please.

-----Thank you for your comment. The sentence was improved.

* Lines 59, 60: not clear: check the sentence, please.

-----Thank you for your comment. The sentence was improved.

* Line 79: acronym (SP) should be explicited to help Readers, even if obvious + Cite the type of SP and the producer...

----This is a good point. The abbreviation of water reducing agent was deleted and the full name of water reducing agent was applied.

* Lines 82,83: How did you determine the particle size distribution? With a laser granulometer? If yes, please mention it and the model you used...

----This is a very good point. The particle passing percentage was determined by the test sieving method for fineness of cement according to Chinese Standard GB/T 1345-2005).

* Table 1: what are SL and SF??? You never cited them before! Check, please.

-----Thank you for your comment. SL and SF were written errors and were corrected.

* Line 97: Please detail: XRD instrument used and experimental conditions. Idem for SEM observations: did you dry the samples? metallised them?

----This is a very good point. XRD instrument used and experimental conditions were added in Part 2.3. All samples were dried, specimens for SEM observations were metallised, while specimens for XRD instrument were not metallised. 

* Line 102: which is the viscometer manufacturer, even if obvious? Which spindle did you use?

-----Thank you for your suggestion. The detailed information for viscometer was added in Part 2.3.

* Line 104: which is the  manufacturer of TH2810D instrument?

----This is a very good point. The manufacturer of TH2810D instrument was added in Part 2.3.

* Line 121: what was the shear rate for plastic viscosity determination?

-----Thank you for your suggestion. The shear rate for plastic viscosity determination was 8 rpm.

* Line 144: How can there be a polarization resistance if you worked with an AC voltage at 10 kHz? Check, please. It may be due to the fact that RHA is a pozzolanic material and you have to let first portlandite to form and then to react with RHA. Thus, setting time and hardening are delayed with the increase in RHA content.

-----Thank you for your suggestion. All mistakes were corrected.

* Lines 150-152: As previously written: how can there be a polarization resistance if you worked with an AC voltage at 10 kHz? Did you use Vicat needle to determine setting time? If not, you should determine it.

----This is a very good point. The mistake was corrected. Just as suggested the main reason was the influence of RHA on the cement setting.

* Line 198: Please, mention the JCPDS card numbers you used for XRD peaks indexation.

-----Thank you for your suggestion. JCPDS card numbers used for XRD peaks indexation were added.

* Line 201: How can C4AF content, which is due to OPC, increase when RHA content increases? Check, please.

----This is a very good point. The hydration products determined by X-ray diffraction were checked and corrected.

* Lines 201-203: C3S decrease with increasing addition of RHA  is due  to the fact that you have 15 wt% less cement in RHA4 sample respect to RHA1...

-----Thank you for your suggestion. The errors were corrected just as suggested.

* Figure 10: Is the non-indexed peak below 30° due to CSH? Please, check

----This is a very good comment. The results were checked and errors were corrected.

* Line 222: what does it mean that the microstructure improves? This is not clear.

----This is a very good point. It could be observed from scanning electron microscope (SEM) photos RHA was able to improve the compactability of cement paste.

* Line 222: the decrease of cracking is rather surprising, as usually, RHA increases cracking: Ameri et al., Optimum rice husk ash content and bacterial concentration in self-compacting concrete, Construction and Building Materials 222 (2019) 796–813. You should explain this.

----This is a very good comment. However, previous research pointed out that the amorphous silica in RHA reacted with calcium hydroxide formed C-S-H. Moreover, the particle size of RHA was less than 45 µm on average which operated a refinement on the pore structure, acts as nucleation point for hydration products. Therefore, RHA was able to improve the compactability of cement paste.

* Figure 11: are compact parts in RHA8 and RHA12 samples RHA grains?

----This is a very good point. However, the particle size of rice husk ash is smaller than that of cement which operated a refinement on the pore structure, acted as nucleation point for hydration products. Therefore, RHA was able to improve the compactability of cement paste as described in Ref [33]. More research will be carried out to confirm that RHA is able to improve the compactability of cement paste in the future.

* Line 246: How can the sample polarize when doing AC measurements?

----This is a very good point. The errors were corrected.

* Line 250: what does it mean that the microstructure improves? This is not clear.

+ see the attached pdf file for other minor revisions.

----This is a very good comment. In this paper in means that RHA was able to improve the compactability of cement paste. All errors were corrected.

Round 2

Reviewer 2 Report

in my opinion: the article after revisions is suitable and can be accepted in this form

Author Response

Comments and Suggestions for Authors in my opinion: the article after revisions is suitable and can be accepted in this form

-----Thank you for your comment.  This paper has been checked overally again.

Reviewer 3 Report

Dear Authors,

thank you for having accepetd almost all suggested corrections. However, in my opinion, there are still some points to change before final acceptance:

* Lines 41-44: the sentence is not clear, check, please,

* Lines 62-66: "et al." used twice in these sentences is not correct... Check please. Maybe "etc" is more correct?

* Line 91 + Table 1: I am sorry, I don't know this Chinese standard but, I have some serious doubts about the existence of sieves with the apertures reported in Table 1. Please, check.

* Table 2: If we sum all the oxides the total is much higher than 100% (116.77%), please check these results...

* Lines 168-170: Check this sentence, it is not coherent with what previously written (lines 161-164) and what written just after (lines 170-171)...

* Lines 222-223: This is a strange result: as already said to you, if RHA is a pozzolanic product, it should react with CH and you should observe a decrease of its concentration and not an increase of CH concentration with time. How were these samples cured? It is not clear in the experimental part. Please, add also this information.

* Lines 235, 236: Why do XRD patterns showed the exact opposite result??? Check the sentence, please.

* Line 270: as said before, check this point, please.

+ see the attached pdf file for other minor corrections.

Best regards.

Reviewer 2

Author Response

* Lines 41-44: the sentence is not clear, check, please,

-----Thank you for your comment. This sentence had been deleted.

* Lines 62-66: "et al." used twice in these sentences is not correct... Check please. Maybe "etc" is more correct?

-----This is a very good point. "et al." had been corrected to “etc”.

* Line 91 + Table 1: I am sorry, I don't know this Chinese standard but, I have some serious doubts about the existence of sieves with the apertures reported in Table 1. Please, check.

-----This is a good point. However, the particle passing percentage of cementitious materials is provided by Yancheng Conch Cement Co., Ltd. The method for testing the particle passing percentage of cementitious materials is a secret of scientific research.

* Table 2: If we sum all the oxides the total is much higher than 100% (116.77%), please check these results...

-----Thank you for your comment. Table 2 was checked and errors were corrected.

* Lines 168-170: Check this sentence, it is not coherent with what previously written (lines 161-164) and what written just after (lines 170-171)...

-----This is a good point. This sentence was checked and errors were corrected.

* Lines 222-223: This is a strange result: as already said to you, if RHA is a pozzolanic product, it should react with CH and you should observe a decrease of its concentration and not an increase of CH concentration with time. How were these samples cured? It is not clear in the experimental part. Please, add also this information.

-----Thank you for your comment. This strange result had been checked carefully, errors were corrected.

* Lines 235, 236: Why do XRD patterns showed the exact opposite result??? Check the sentence, please.

-----This is a very good point. Lines 235, 236 were written errors and was checked and corrected carefully by all authors.

* Line 270: as said before, check this point, please.

-----Thank you for your suggestion. Line 270 was checked and revised.

+ see the attached pdf file for other minor corrections.

-----Thank you for your comment. All errors were corrected as suggested. The modification parts were marked red font.